# CCR7 defines a precursor for murine iNKT cells in thymus and periphery

Haiguang Wang, Kristin A Hogquist*

The Department of Laboratory Medicine and Pathology, Center for Immunology, University of Minnesota, Minneapolis, United States

**Abstract** The precise steps of iNKT subset differentiation in the thymus and periphery have been controversial. We demonstrate here that the small proportion of thymic iNKT and mucosal associated invariant T cells that express CCR7 represent a multi-potent progenitor pool that gives rise to effector subsets within the thymus. Using intra-thymic labeling, we also showed that CCR7$^+$ iNKT cells emigrate from the thymus in a *Klf2* dependent manner, and undergo further maturation after reaching the periphery. *Ccr7* deficiency impaired differentiation of iNKT effector subsets and localization to the medulla. Parabiosis and intra-thymic transfer showed that thymic NKT1 and NKT17 were resident—they were not derived from and did not contribute to the peripheral pool. Finally, each thymic iNKT effector subset produces distinct factors that influence T cell development. Our findings demonstrate how the thymus is both a source of iNKT progenitors and a unique site of tissue dependent effector cell differentiation.

DOI: https://doi.org/10.7554/eLife.34793.001

## Introduction

Invariant natural killer T (iNKT) cells are αβ T cells expressing semi-invariant T cell receptors (TCR) that respond to self-lipids or foreign-lipids presented by the MHC-I like antigen-presenting molecule CD1d (*Bendelac et al., 2007*). They are numerically less abundant than conventional peptide recognizing CD4 and CD8 T cells but profoundly important, secreting a variety of cytokines early after stimulation or in the steady state to influence immune responses and tissue homeostasis (*Lee et al., 2013*; *Lynch et al., 2012*; *Lee et al., 2015*; *Engel and Kronenberg, 2014*). Previous work has demonstrated that despite being monospecific, iNKT cells display substantial functional heterogeneity (*Coquet et al., 2008*; *Doisne et al., 2009*; *Doisne et al., 2011*; *Havenar-Daughton et al., 2012*; *Michel et al., 2007*; *Terashima et al., 2008*; *Watarai et al., 2012*), being composed of three predominant effector subsets, NKT1, NKT2 and NKT17 according to the expression of key transcription factors and cytokine production (*Lee et al., 2013*; *Gapin, 2016*). For instance, NKT1 cells are T-bet$^+$ PLZF$^{low}$ and produce IFN-γ; NKT17 cells are ROR-γt$^+$ PLZF$^{int}$ and produce IL-17; while NKT2 cells are PLZF$^{high}$, and produce IL-4 (*Lee et al., 2013*; *Gapin, 2016*). Through the selective activation and distinct localization of various iNKT effector subsets, iNKT cells can modulate immune responses and tissue homeostasis in different fashions (*Lynch et al., 2012*; *Lee et al., 2015*; *Lynch et al., 2016*). iNKT cells are positively selected in the thymus, in response to agonist interactions with self-lipid/ CD1d expressing double positive thymocytes (DP) in the thymic cortex (*Egawa et al., 2005*; *Gapin et al., 2001*; *Moran et al., 2011*). At this point the cells express a high level of CD24, and are considered the most immature, or 'stage 0'. In contrast, matured functionally competent effector iNKT cells localize predominantly in the thymic medulla (*Lee et al., 2015*), and thymic medullary epithelial cells impact their functional maturation (*White et al., 2014*). Nonetheless, the developmental steps from stage 0 iNKT cells to mature iNKT effector subsets have remained unclear. Previous work used CD44 and NK1.1 to define 'stages' of iNKT development. However, more recent work showed that these markers are heterogeneously expressed on the mature cytokine producing iNKT effector

*For correspondence:
hogqu001@umn.edu

subsets (*Coquet et al., 2008*; *Doisne et al., 2009*; *Doisne et al., 2011*; *Havenar-Daughton et al., 2012*; *Michel et al., 2007*; *Terashima et al., 2008*; *Watarai et al., 2012*). Therefore, the field lacks precise identification of an iNKT multipotent precursor cell. Previous work in our lab, using T-bet and IL-4 reporter mice, identified T-bet$^+$ NKT1, ROR-γt$^+$ NKT17, and IL-4$^+$ PLZF$^{hi}$ NKT2 cells as terminally differentiated (*Lee et al., 2013*). However, IL-4$^-$ PLZF$^{hi}$ iNKT cells still retained precursor potential, and could convert to T-bet$^+$ NKT1 cells in the thymus (*Lee et al., 2013*). These results suggest there is heterogeneity within the PLZF$^{hi}$ iNKT cell population, and led us to hypothesize that there exists a population of progenitor cells within the PLZF$^{hi}$ iNKT cells. Indeed, transcriptional profiling by RNA-Seq indicated IL-4$^-$ PLZF$^{hi}$ iNKT cells had low similarity with the three effector subsets, including IL-4 producing PLZF$^{hi}$ iNKT cells (*Lee et al., 2016*). Single cell RNA-Seq analysis from another group also suggested there might be a progenitor population within the PLZF$^{hi}$ iNKT cells (*Engel et al., 2016*).

Here we identify a previously unknown sub-population of PLZF$^{hi}$ CCR7$^+$ iNKT cells as precursors for all three iNKT effector subsets. CCR7$^+$ iNKT cells efficiently emigrate from the thymus in a *Klf2* dependent manner and then undergo further maturation on site. However, some iNKT cells maintain residency in the thymus where they undergo differentiation without circulating. The thymic and peripheral pools of iNKT effector subsets do not exchange and therefore depend on CCR7$^+$ iNKT cells for their establishment. In addition to marking the precursor pool, CCR7 also directs iNKT progenitor cells to localize to the thymic medulla and is required for differentiation of iNKT effector subsets. We further establish that thymic iNKT cells influence T cell development and thymic tissue homeostasis.

## Results

### CCR7$^+$ iNKT and MAIT cells are at an early stage of development and represent a precursor pool for effector subsets in the thymus

To identify iNKT cells at an early stage of development in the thymus, we used mice that express green fluorescent protein (GFP) under the control of the recombination-activating gene 2 (*Rag2*) promoter via a bacterial artificial chromosome transgene (*Rag2*$^{GFP}$), in which GFP expression indicates the 'age' of cells (*Boursalian et al., 2004*; *McCaughtry et al., 2007*). We found most thymic iNKT cells were *Rag2*$^{GFP}$ negative, while a small proportion of iNKT cells express *Rag2*$^{GFP}$ (*Figure 1A*). The iNKT cells with the highest GFP also express CD24 and high level of CD69, indicating they are stage 0 iNKT cells that are immediate after agonist selection (*Figure 1A*). Since we identified CCR7 as a potential progenitor specific gene (*Lee et al., 2016*), and a small population of CCR7$^+$ iNKT cells could be specifically detected (*Figure 1—figure supplement 1A*), we further examined CCR7 expression in iNKT cells in *Rag2*$^{GFP}$ mice. CCR7$^+$ iNKT cells expressed high to intermediate amounts of GFP, suggesting they are immature and recently derived from stage 0 iNKT cells (*Figure 1A* and *Figure 1—figure supplement 1A*). Whereas, the *Rag2*$^{GFP}$ lowest cells (most mature) did not express CCR7 (*Figure 1A*). According to the conventional 'staging' system of iNKT cells in B6 mice, CCR7$^+$ iNKT cells were predominantly NK1.1$^-$ CD44$^{lo}$ (stage 1) but also included some NK1.1$^-$ CD44$^+$ cells (stage 2) (*Figure 1B*), although most NK1.1$^-$ CD44$^+$ (stage 2) cells were not CCR7$^+$ as this gate mostly included functionally mature NKT2 and NKT17 cells, and NKT1 cells were NK1.1$^+$ CD44$^+$ (stage 3) (*Figure 1B*). Furthermore, most CCR7$^+$ iNKT cells did not express T-bet, ROR-γt, IL-4 (human CD2) (*Figure 1C*). Rather, CCR7$^+$ iNKT cells expressed abundant PLZF and LEF1 (*Figure 1C*), which are both essential for iNKT cells development and proliferation (*Carr et al., 2015*; *Savage et al., 2008*; *Kovalovsky et al., 2008*).

To confirm that the CCR7$^+$ iNKT cells were at an early developmental stage, we sought to track a 'wave' of developing iNKT cells using busulfan induced bone marrow chimeras (*Figure 1—figure supplement 2A*). We showed that, within CD45.1$^+$ donor derive CD1d tetramer$^+$ iNKT cells, the immature CD24$^+$ CD44$^-$ stage 0 iNKT cells were enriched at an early time point (4 weeks) and contracted at a later time point (5 weeks), while the NK1.1$^+$ CD44$^+$ mature iNKT cells were scarce at 4 weeks but abundant at 5 weeks (*Figure 1—figure supplement 2B*), suggesting this approach tracks the developmental steps of iNKT cells. With this approach, CCR7$^+$ iNKT cells (with lower CD44 and T-bet) were abundant at the early time point (4 weeks) after bone marrow introduction and

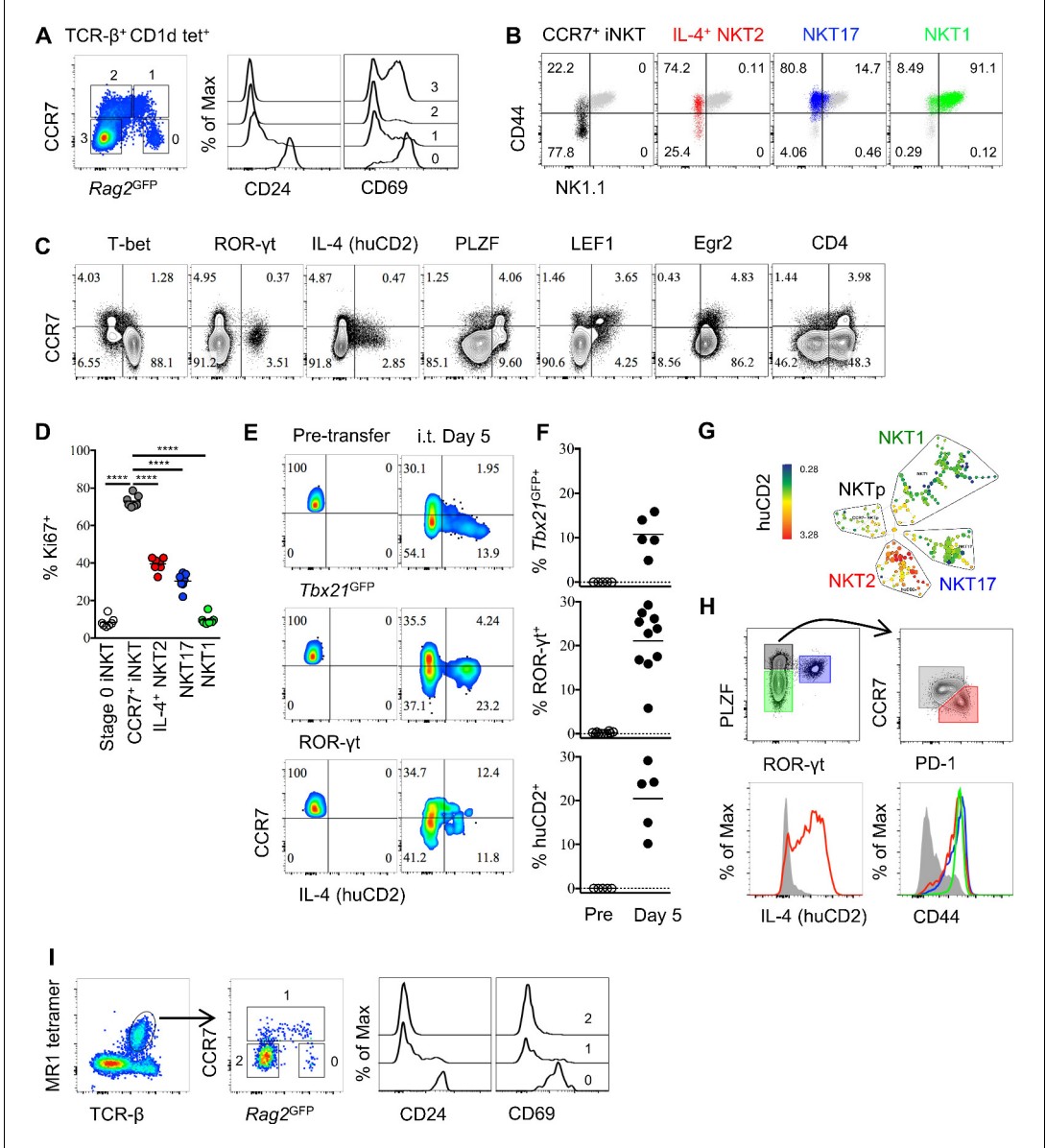

**Figure 1.** Thymic CCR7[+] iNKT and MAIT cells are at an early developmental stage and give rise to distinct effector subsets in the thymus. (**A**) Expression of *Rag2*[GFP] and CCR7 in thymic iNKT cells (TCR-β[+] CD1d-PBS57[+]) (left column), and of CD24 (middle column) and CD69 (right column) on cells with different levels of *Rag2*[GFP]. Data are representative of 3 independent experiments with 3–4 mice in each. (**B**) Expression of NK1.1 and CD44 in thymic CCR7[+] iNKT (black dots), IL-4[+] (human CD2[+]) NKT2 (red dots), ROR-γt[+] NKT17 (blue dots), T-bet[+] NKT1 (green dots) cells together with total thymic iNKT cells (grey dots). Numbers in quadrants indicate percent cells in each for CCR7[+] iNKT (black dots), IL-4[+] (human CD2[+]) NKT2 (red dots), ROR-γt[+] NKT17 (blue dots) and T-bet[+] NKT1 (green dots) cells. Data are representative of 4 independent experiments with 2–3 mice in each. (**C**) Expression of T-bet, ROR-γt, huCD2, PLZF, LEF1, Egr2 and CD4 with CCR7 in thymic iNKT cells (TCR-β[+] CD1d-PBS57[+] CD24[−]). Data are representative of 3 independent experiments with 2–3 mice in each. Numbers in quadrants indicate percent cells in each (throughout). (**D**) Frequency of Ki67[+] cells in each population of thymic iNKT cells. Data are pooled from three independent experiments with 2–3 mice in each. [****]*p*<0.0001 (one-way ANOVA, Tukey's multiple comparisons test) Each symbol represents an individual mouse; small horizontal lines indicate the mean. (**E**) Expression of *Tbx21*[GFP], ROR-γt and human CD2 in CCR7[+] iNKT cells sorted from BALB/c *Tbx21*[GFP] KN2 mice before intra-thymic transfer (left column) or 5 days after transfer in the thymus of congenic BALB/c recipient mice (right column). (**F**) Frequency of *Tbx21*[GFP+], ROR-γt[+] or human CD2[+] cells in donor cells before or 5 days after intra-thymic transfer into the thymus of congenic BALB/c recipient mice. Each symbol represents an individual recipient mouse; small horizontal lines indicate the mean. (**G**) SPADE analysis of thymic iNKT cells from B6 KN2 mice supports that CCR7[+] NKTp are a distinct lineage from the effector subsets, NKT1, NKT2 and NKT17. Representative figure shows differential expression of human CD2 in each population of iNKT cells. (**H**) CCR7 and PD-1 distinguish two cell populations (top row, right column) within PLZF[hi] iNKT cells (top row, left column), and expression of human CD2 and CD44 in CCR7[+] NKTp (grey), NKT1 (green), NKT2 (red) and NKT17 (blue) are shown as overlays (bottom row). Data are representative of 3 independent experiments with 3 mice in each. (**I**) Expression of *Rag2*[GFP] together with CCR7 (middle column) in thymic MAIT cells (far left column), and expression of

*Figure 1 continued on next page*

*Figure 1 continued*

CD24 and CD69 in CCR7$^+$ and CCR7$^-$ MAIT cells with different level of *Rag2*$^{GFP}$ (far right two columns). Data are representative of 2 independent experiments with 3 mice in each.

DOI: https://doi.org/10.7554/eLife.34793.002

The following source data and figure supplements are available for figure 1:

**Source data 1.** High proliferative and precursor potential of CCR7$^+$ iNKT cells.

DOI: https://doi.org/10.7554/eLife.34793.006

**Source data 2.** Thymic CCR7$^+$ iNKT cells are distinguished from stage 0 iNKT cells and give rise to iNKT subsets in periphery.

DOI: https://doi.org/10.7554/eLife.34793.007

**Source data 3.** Consistent, robust and unbiased labeling of thymocytes by intra-thymic injection of biotin.

DOI: https://doi.org/10.7554/eLife.34793.008

**Figure supplement 1.** Specific CCR7 staining in iNKT cells and gating strategy of iNKT subsets and CD4/CD8 profile of CCR7$^+$ iNKT and CCR7$^+$ MAIT cells.

DOI: https://doi.org/10.7554/eLife.34793.003

**Figure supplement 2.** Thymic CCR7$^+$ iNKT cells are enriched at an early timepoint in busulfan induced BM chimera.

DOI: https://doi.org/10.7554/eLife.34793.004

**Figure supplement 3.** Ultrasound imaging guided intra-thymic injection.

DOI: https://doi.org/10.7554/eLife.34793.005

decreased at the later time point (5 weeks) (with increased CD44 and T-bet) (*Figure 1—figure supplement 2C*).

As CCR7$^+$ iNKT cells expressed a high level of LEF1 (*Figure 1C*), a transcription factor that is essential for iNKT cells proliferation, we examined Ki67 expression. Most CCR7$^+$ iNKT cells expressed Ki67 (>75%) compared to the three effector subsets or the stage 0 iNKT cells (*Figure 1D*, *Figure 1—figure supplement 1B,C*), suggesting they are highly proliferative. Stage 0 iNKT cells received strong TCR signal during agonist selection which could be indicated by the level of *Nr4a1* using *Nr4a1*$^{GFP}$ mice (also known as Nur77$^{GFP}$ mice which will be used throughout the manuscript) (*Moran et al., 2011*). Highly proliferative as CCR7$^+$ iNKT cells are, the Nur77$^{GFP}$ could be diluted were they not continuously receiving TCR stimulation. We showed CCR7$^+$ iNKT cells had much lower level of Nur77$^{GFP}$ than stage 0 iNKT cells (*Figure 1—figure supplement 2D,E*). A previous study suggested that the transcription factor Egr2 controls the development of iNKT cells and is highly expressed in iNKT thymic precursors (*Seiler et al., 2012*). Consistently, we observed a high expression of Egr2 in CCR7$^+$ iNKT cells while most thymic iNKT cells were positive for Egr2 expression (*Figure 1C*). Moreover, consistent with a previous report that iNKT cells go through the CD4$^+$ stage after stage 0 iNKT cells (*Benlagha et al., 2005*), we showed CCR7$^+$ iNKT cells indeed exclusively expressed CD4, but not CD8 (*Figure 1C*, *Figure 1—figure supplement 1D*).

These data raise the possibility that CCR7$^+$ iNKT cells serve as progenitors for differentiation into effector subsets. We then directly assessed the potential of CCR7$^+$ iNKT cells to develop into iNKT effector subsets by sorting CCR7$^+$ thymocytes (CD4$^+$ *Tbx21*$^{GFP-}$ huCD2$^-$ CD24$^-$ CD8$^-$) from *Tbx21*$^{GFP}$/KN2 mice and injecting them intra-thymically into congenic hosts. We used an ultrasound guided imaging technique to obtain accurate and efficient intra-thymic injection (*Figure 1—figure supplement 3A*). Five days after intra-thymic transfer, all three effector subsets, NKT1, NKT2 and NKT17 cells were detected (*Figure 1E,F*). To test the precursor potential of thymic CCR7$^+$ iNKT cells in the periphery, we also injected the sorted CCR7$^+$ thymocytes intravenously into congenic hosts and detected substantial differentiation of NKT1, NKT2 and NKT17 cells in the spleen (*Figure 1—figure supplement 2F,G*), demonstrating that iNKT progenitors do not require the thymus for effector differentiation.

To better identify markers that can distinguish the precursor cells from cytokine producing NKT2 effector cells, we performed Spanning-tree Progression Analysis of Density-normalized Events (SPADE analysis) on total thymic iNKT cells based on a range of key transcription factors, surface markers and cytokines. Importantly, CCR7$^+$ iNKT cells were clearly separated from the three major effector subsets, (*Figure 1G*, *Figure 1—figure supplement 2H*). Within the PLZF$^{hi}$ iNKT cells, CCR7 and PD-1 best distinguished progenitors from effectors, and only PLZF$^{hi}$ PD-1$^+$ iNKT cells produced IL-4 (human CD2$^+$) in the steady state (*Figure 1H*).

Previous reports suggested that the two lineages of innate-like T cells—iNKT cells and mucosal associated invariant T (MAIT) cells—have similarities in aspects of their phenotype and functions, such that developmental parallels might exist between the two (*Koay et al., 2016*; *Wang and Hogquist, 2016*). Indeed, similar to CCR7[+] iNKT cells (*Figure 1A*), CCR7[+] MAIT cells also had high to intermediate *Rag2*[GFP] expression while being low for CD24 and CD69 expression, suggesting they are also at an early stage of development after agonist selection (*Figure 1I*). Mature MAIT cells are composed of only two distinct effector subpopulations, ROR-γt[+] and T-bet[+] MAIT cells (*Koay et al., 2016*). CCR7[+] MAIT cells did not express T-bet or ROR-γt (*Figure 1—figure supplement 1E*). Together, these data suggest that both CCR7[+] iNKT cells and CCR7[+] MAIT cells are progenitor populations that give rise to their corresponding iNKT and MAIT effector cells.

## CCR7[+] iNKT cells are the predominant emigrating iNKT cells and they depend on *Klf2* for thymic emigration

To investigate the phenotype of iNKT cells emigrating from the thymus, we performed intra-thymic injection of a biotinylating agent (NHS-biotin) to label thymocytes (*Figure 1—figure supplement 3A*) and analyze peripheral lymphoid organs 24 hr later (*Figure 2A*). This technique showed robust and unbiased labeling of nearly 50% of all thymocytes (*Figure 1—figure supplement 3B,C*) and did not interfere with the specificity of CD1d tetramer staining (*Figure 1—figure supplement 3D*). Due to the low frequency of recent thymic emigrants (RTE) amongst total peripheral T lymphocytes (*Boursalian et al., 2004*; *McCaughtry et al., 2007*), we performed magnetic enrichment of biotin[+] cells in the spleen. These two techniques combined offers a tool to accurately detect RTEs in periphery, as biotin[+] splenic CD4[+] or CD8[+] T cells are predominantly *Rag2*[GFP+] (*Figure 2—figure supplement 1A,B*). Using this tool, recent thymic emigrant iNKT cells (called 'RTE iNKT') could be readily detected (*Figure 2A*), while biotin[−] iNKT cells within the unbound fraction are largely (>97%) 'old' iNKT cells—those cells already residing in the periphery before intra-thymic labeling (*Figure 2A*). As expected, the RTE iNKT cells have higher *Rag2*[GFP] compared to biotin[−] iNKT cells (*Figure 2—figure supplement 1C*). We further found that RTE iNKT cells were highly enriched for CCR7[+] cells compared to thymic or biotin[−] iNKT cells (*Figure 2B*). Moreover, within the RTE iNKT cells, the CCR7[+] cells expressed higher *Rag2*[GFP] than CCR7[−] cells (*Figure 2—figure supplement 1D*). In agreement with these data, *Rag2*[GFP+] splenic iNKT cells were predominantly CCR7[+], while most *Rag2*[GFP−] splenic iNKT cells did not express CCR7 (*Figure 2—figure supplement 1E*).

The transcription factor Kruppel-like factor 2 (*Klf2*) is crucial for the thymic egress of conventional CD4 and CD8 lymphocytes (*Carlson et al., 2006*). However, its role in thymic emigration of iNKT cells is currently unknown. To test whether thymic egress of iNKT cells was dependent on *Klf2*, we analyzed mixed bone-marrow chimeras with wild type and *Klf2*-deficient bone marrow. We performed intra-thymic labeling to track RTE. As expected, *Klf2*-deficient conventional CD4 and CD8 T cells were enriched in the thymus and underrepresented amongst peripheral cells (*Figure 2C,D*). Similarly, *Klf2*-deficient iNKT cells were also enriched in the thymus and underrepresented amongst RTE and biotin[−] iNKT cells populations (*Figure 2C,D*), suggesting thymic emigration of iNKT cells is dependent on *Klf2*. As *Klf2* regulates expression of S1PR1 (*Carlson et al., 2006*), this is consistent with the previously reported requirement for S1PR1 in iNKT cell thymic egress (*Allende et al., 2008*). Next, we sought to assess *Klf2* expression in thymic and RTE iNKT cells using *Klf2*[GFP] reporter mice. The majority of thymic iNKT cells did not express *Klf2*, though thymic CCR7[+] iNKT cells or RTE iNKT cells highly expressed *Klf2* comparable to mature thymic CD8 single positive (SP) thymocytes or RTE CD8 T cells, respectively (*Figure 2E*, *Figure 2—figure supplement 2A*). Altogether, these results suggest that CCR7[+] iNKT cells are the predominant thymic emigrating iNKT cell, and they do so in a *Klf2* dependent manner.

## RTE iNKT are functionally immature and undergo further differentiation in the periphery

In further characterizing RTE iNKT cells, we observed that T-bet[+], and ROR-γt[+] cells were substantially underrepresented amongst RTE iNKT cells compared to total thymic iNKT cells (*Figure 3A,B*), suggesting that both T-bet[+] NKT1 and ROR-γt[+] NKT17 cells may be resident in the thymus. This is consistent with the previously reported long-term retention of thymic NK1.1[+] iNKT cells (*Berzins et al., 2006*). Similarly, thymic NKT2 cells that produce high levels of IL-4 (human CD2)

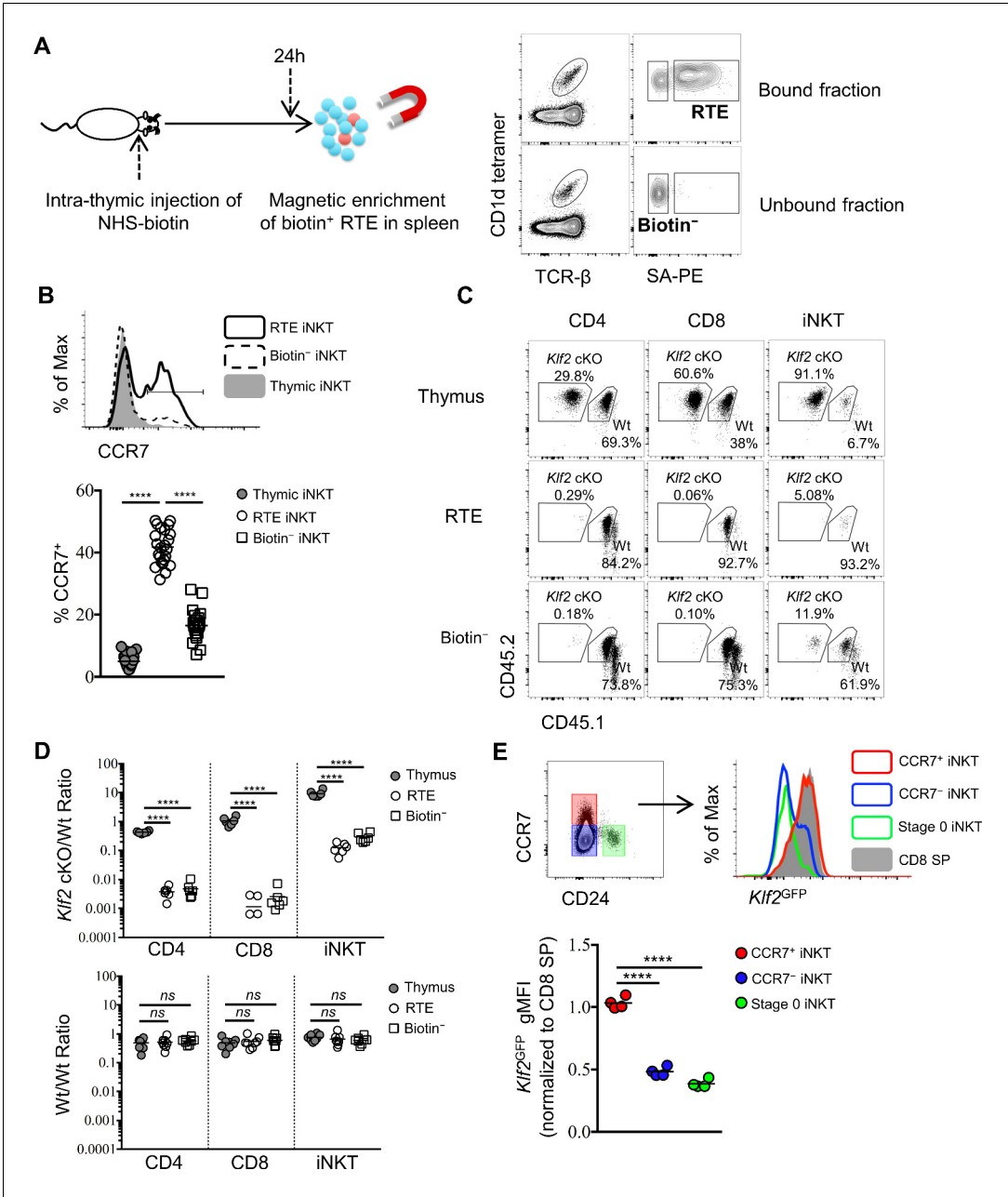

**Figure 2.** CCR7[+] iNKT cells are enriched in the emigrating iNKT population and depend on *Klf2*. (**A**) Experimental scheme to track recent thymic emigrants (RTE) through intra-thymic injection of NHS-biotin (left column) followed by enrichment of biotin[+] splenic cells 24 hr after intra-thymic labeling, compared to 'old' cells (biotin[−] cells from spleen) (right column). (**B**) Expression of CCR7 in thymic, RTE and biotin[−] iNKT cells, the grey shade represents thymic iNKT cells, the solid black line represents RTE iNKT cells, the dashed black line represents biotin[−] iNKT cells (top row); frequency of CCR7[+] cells in thymic, RTE and biotin[−] iNKT cells, the solid grey circle represents thymic iNKT cells, the open circle represents RTE iNKT cells, the open square represents biotin[−] iNKT cells (bottom row). Data are pooled from 5 independent experiments with 3–6 mice in each. [****]*p*<0.0001 (one-way ANOVA) Each symbol represents an individual mouse; small horizontal lines indicate the mean. (**C**) Mixed bone marrow chimeras were generated with 25:75 ratio of donor bone marrow cells using CD45.2[+] CD45.2[+] B6 *Klf2* cKO cells and CD45.1[+] CD45.2[+] B6 Wt cells, or with 50:50 ratio of donor bone marrow cells using CD45.2[+] CD45.2[+] B6 Wt cells and CD45.1[+] CD45.2[+] B6 Wt cells. Eight weeks later, chimeras received intra-thymic labeling with NHS-biotin to track RTE among CD4, CD8 and iNKT cells. Data are representative of 2 independent experiments with 4–8 mice in each. (**D**) Statistical analysis of thymic, RTE and biotin[−] CD4 T cells, CD8 T cells and iNKT cells in 8-week-old BM chimeras reconstituted with *Klf2* cKO and Wt cells (top row) or with Wt and Wt cells (bottom row).

*Figure 2 continued on next page*

*Figure 2 continued*

Data are pooled form 2 independent experiments with 4–8 mice in each. Numbers indicate the ratio between the cells derived from each donor source. [****]$p<0.0001$ (one-way ANOVA). *ns*, not significant, $p>0.05$ (one-way ANOVA). Each symbol represents an individual mouse; small horizontal lines indicate the mean. (E) Expression of $Klf2^{GFP}$ in CCR7$^{+}$ iNKT, stage 0 iNKT, CCR7$^{-}$ iNKT cells and CD8 SP cells (top row). Normalized gMFI of $Klf2^{GFP}$ in CCR7$^{+}$ iNKT, stage 0 iNKT, CCR7$^{-}$ iNKT cells (bottom row). Data are pooled from 2 independent experiments with 2 mice in each. [****]$p<0.0001$ (one-way ANOVA). Each symbol represents an individual mouse; small horizontal lines indicate the mean.

DOI: https://doi.org/10.7554/eLife.34793.009

The following source data and figure supplements are available for figure 2:

**Source data 1.** RTE iNKT cells are CCR7$^{+}$ and depends on *Klf2* for emigration.

DOI: https://doi.org/10.7554/eLife.34793.012

**Source data 2.** RTE iNKT and CCR7$^{+}$ RTE iNKT cells are *Rag2*GFP$^{+}$.

DOI: https://doi.org/10.7554/eLife.34793.013

**Source data 3.** High $Klf2^{GFP}$ in RTE iNKT cells.

DOI: https://doi.org/10.7554/eLife.34793.014

**Figure supplement 1.** Intra-thymic labeling with NHS-biotin to identify RTEs in the periphery and *Rag2*$^{GFP+}$ splenic iNKT cells are CCR7$^{+}$.

DOI: https://doi.org/10.7554/eLife.34793.010

**Figure supplement 2.** RTE iNKT cells express high level of $Klf2^{GFP}$.

DOI: https://doi.org/10.7554/eLife.34793.011

were also less likely to be found amongst RTE (*Figure 3B*). Though CCR7$^{+}$ iNKT cells were predominant within the RTE population, a small proportion of T-bet$^{+}$ NKT1 cells were still observed (*Figure 3A,B*). However, cells within this small population failed to express Qa2, while the thymic T-bet$^{-}$ NKT1 and splenic biotin$^{-}$ T-bet$^{+}$ NKT1 cells both expressed high level of Qa2 (*Figure 3C,D*). As Qa2 has commonly been used to mark the most mature thymocytes, this suggests that RTE NKT1 cells were recently derived from progenitors, as opposed to emigrating as a mature cell. Indeed, in cells collected 72 hr after intra-thymic labeling, we observed a progressive increase in Qa2 in RTE NKT1 cells (*Figure 3C,D*). Furthermore, RTE T-bet$^{+}$ NKT1 expressed significantly higher PLZF at 24 hr, while the expression was down regulated over 72 hr (*Figure 3D*). Taken together, these results strongly suggested that RTE T-bet$^{+}$ NKT1 cells were not directly derived from mature thymic NKT1 cells, but more likely recently differentiated from precursors. To further support this hypothesis, we sorted CCR7$^{+}$ thymocytes (CD4$^{+}$ *Tbx21*$^{GFP-}$ huCD2$^{-}$ CD24$^{-}$ CD8$^{-}$) from *Tbx21*$^{GFP}$/KN2 mice and injected them intra-thymically into congenic hosts (*Figure 3—figure supplement 1A*). Five days later, all three effector subsets, NKT1, NKT2 and NKT17 cells were detected in the spleen (*Figure 3—figure supplement 1B,C*), indicating CCR7$^{+}$ iNKT cells rapidly emigrate to the periphery and serve as precursors that give rise to iNKT effector subsets.

## *Ccr7* deficiency impairs differentiation of iNKT effector subsets and their localization to the thymic medulla

As CCR7$^{+}$ iNKT cells serve as progenitors for all three subsets of iNKT cells in the thymus, we sought to evaluate whether CCR7 itself might regulate the differentiation of thymic iNKT cells. To test this, we created mixed bone-marrow chimeras with wild type and *Ccr7*-deficient bone marrow cells, and analyzed the phenotype of thymic iNKT cells derived from both donors. As expected, the generation of double positive (DP) thymocytes was not affected by the absence of CCR7, while iNKT cells derived from the *Ccr7*-deficient donors showed substantially impaired differentiation of all three iNKT effector subsets (*Figure 4A,B*). Similar to CCR7$^{+}$ iNKT cells, CCR7$^{+}$ MAIT cells were at early stage of development (*Figure 1I*) and distinct from effector MAIT cells (*Figure 1—figure supplement 1E*), which suggested the possibility of a similarly important role of CCR7 in the development of MAIT cells. In mixed bone marrow chimeras with wild type and *Ccr7*-deficient bone marrow cells, MAIT cell differentiation was also severely impaired when lacking cell-intrinsic CCR7 (*Figure 4—figure supplement 1A–C*).

It was recently shown that CCR7 responds to CCL21a produced in the thymic medulla and is essential for migration of thymocytes from the cortex to medulla (*Kozai et al., 2017*). To directly

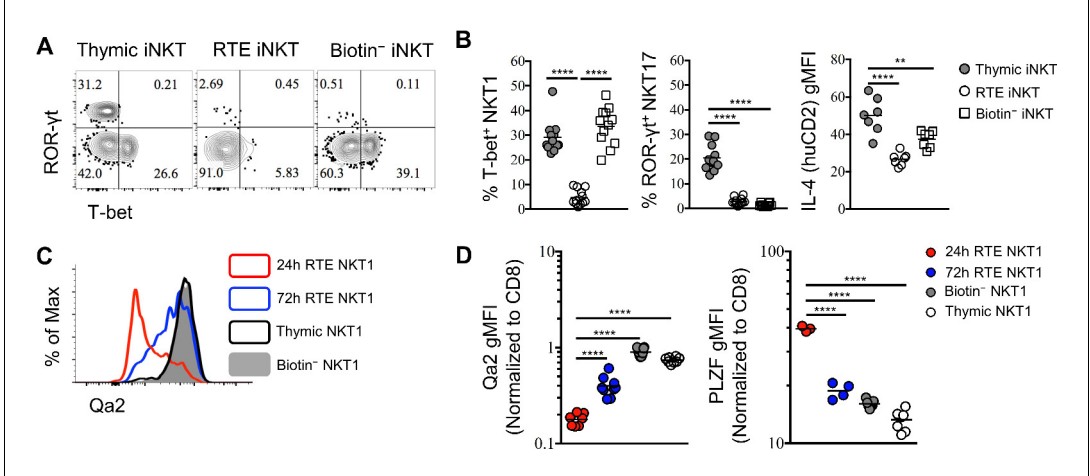

**Figure 3.** RTE iNKT cells are immature and undergo further differentiation after reaching periphery. (A) Expression of T-bet and ROR-γt in thymic, RTE and splenic biotin⁻ iNKT cells. Number in quadrants indicates percent cells in each (throughout). Data are representative of 3 independent experiments with 2–6 mice in each. (B) Frequency of T-bet⁺ NKT1, ROR-γt⁺ NKT17 cells and normalized gMFI of IL-4 (human CD2) in thymic, RTE and splenic biotin⁻ iNKT cells. The solid grey circle represents thymic iNKT cell, the open circle represents RTE iNKT cells, the open square represents biotin⁻ iNKT cells. **p=0.0048, ****p<0.0001 (one-way ANOVA). Data are pooled from 3 independent experiments with 2–6 mice in each. Each symbol represents an individual mouse; small horizontal lines indicate the mean. (C) Expression of Qa2 in thymic, 24 hr RTE, 72 hr RTE and splenic biotin⁻ T-bet⁺ NKT1 cells. Data are representative of 3 independent experiments with 2–3 mice in each. (D) Normalized gMFI of Qa2 in thymic, 24 hr RTE, 72 hr RTE and biotin⁻ T-bet⁺ NKT1 cells (left column). Normalized gMFI of PLZF in thymic, 24 hr RTE, 72 hr RTE and splenic biotin⁻ T-bet⁺ NKT1 cells (right column). ****p<0.0001 (one-way ANOVA). Data are pooled from 3 independent experiments with 2–3 mice in each. Each symbol represents an individual mouse; small horizontal lines indicate the mean.

DOI: https://doi.org/10.7554/eLife.34793.015

The following source data and figure supplement are available for figure 3:

**Source data 1.** RTE iNKT cells are immature and mature further in periphery.

DOI: https://doi.org/10.7554/eLife.34793.017

**Source data 2.** Thymic CCR7⁺ iNKT cells emigrate to periphery and differentiate into effector subsets.

DOI: https://doi.org/10.7554/eLife.34793.018

**Figure supplement 1.** Thymic CCR7⁺ iNKT emigrate to the periphery and undergo further development into effector subsets.

DOI: https://doi.org/10.7554/eLife.34793.016

assess whether CCR7 directs the medullary localization of thymic iNKT cells, we performed CD1d tetramer based immunofluorescence and histocytometry analysis in the mixed bone-marrow chimeras reconstituted with wild type and *Ccr7*-deficient bone marrow donors. CD1d tetramer based immunofluorescence could specifically detect the CD1d restricted iNKT cells in the thymus (*Figure 4C*). Moreover, *Ccr7*-deficient iNKT cells showed significantly reduced localization in the thymic medulla compared to wild type iNKT cells (*Figure 4D–F*). These data suggest that, CCR7 not only plays an indispensable role in differentiation of thymic iNKT effector subsets, but also directs iNKT cells to localize in thymic medulla.

## Thymic iNKT effector subsets are resident and influence thymic tissue homeostasis

To gain insight into the potential tissue residency of thymic iNKT effector subsets, we generated parabiotic mice, which share blood circulation through surgical joining (*Figure 5A*). Congenically distinct CD45.1⁺ and CD45.2⁺ mice were surgically connected for 30 days. Splenic CD4 or CD8 T cells were present at 50:50 ratios in parabiotic pairs consistent with their known circulation patterns (*Figure 5B,C*). In contrast, ~80% of splenic iNKT cells were of host origin (*Figure 5B,C*), confirming that the majority of splenic iNKT cells are not circulating (*Thomas et al., 2011*). Remarkably, >95% of thymic iNKT cells were of host origin (*Figure 5B,C*). Considering that thymic iNKT effector subsets are negative for *Rag2*ᴳᶠᴾ (*Figure 1A*), these data suggest that iNKT cells in the thymus are largely tissue-resident.

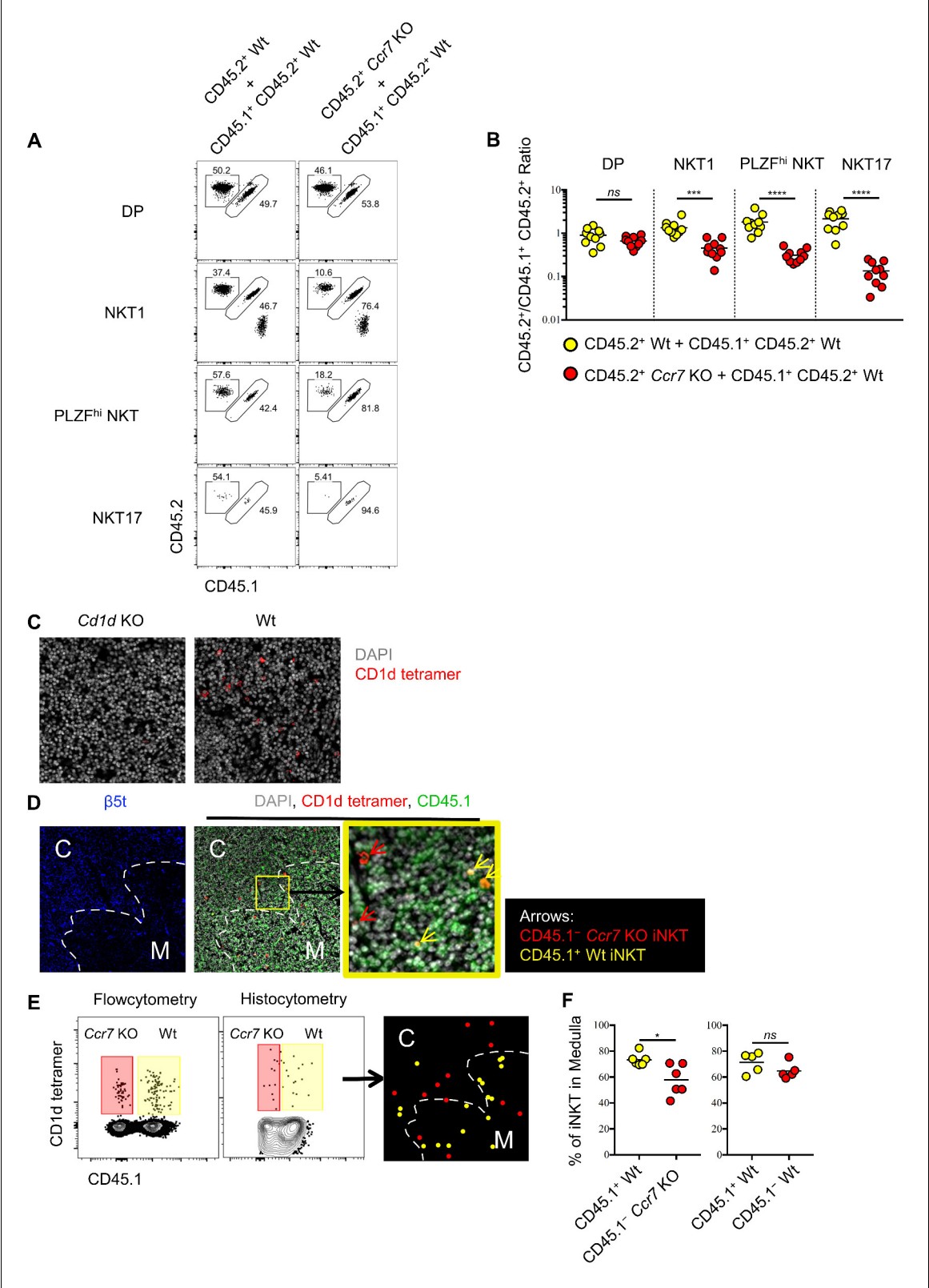

**Figure 4.** *Ccr7* deficiency impairs iNKT subsets differentiation and medullary localization. (**A**) Mixed bone marrow chimeras were generated with 50:50 ratio of donor bone marrow cells using CD45.2+ CD45.2+ B6 *Ccr7* KO cells and CD45.1+ CD45.2+ B6 Wt cells, or with 50:50 ratio of donor bone marrow cells using CD45.2+ CD45.2+ B6 Wt cells and CD45.1+ CD45.2+ B6 Wt cells. Eight weeks later, the frequency of thymic iNKT effector subsets and thymocytes derived from different donor source were analyzed. Data are representative of 3 independent experiments with 4–10 mice in each.

*Figure 4 continued on next page*

*Figure 4 continued*

Numbers adjacent to outlined areas indicate percent cells in each, throughout. (B) Statistical analysis of thymic iNKT subsets and DP thymocytes in 8-week-old BM chimeras reconstituted with *Ccr7* KO and Wt cells or with Wt and Wt cells. Numbers indicate the ratio between the cells derived from each donor source. ***$p$=0.0001 (unpaired two tailed $t$ test). ****$p$<0.0001 (unpaired two tailed $t$ test). *ns*, not significant, $p$>0.05 (unpaired two tailed $t$ test). Data are pooled from 3 independent experiments with 4–10 mice in each. Each symbol represents an individual mouse; small horizontal lines indicate the mean. (C) Thymic sections of *Cd1d* KO (left column) and Wt (right column) B6 mice were stained with CD1d tetramer (red) and the DNA-binding dye DAPI (grey) to visualize iNKT cells *in situ*. (D) Thymic sections of BM chimeras 8 weeks after reconstituted with *Ccr7* KO and Wt cells were stained for the cortical-thymic-epithelial-cell-associated proteasomal subunit β5t (blue), CD1d tetramer (red), CD45.1 (green) and the DNA-binding dye DAPI (grey) to visualize and distinguish cortex, medulla, CD45.1⁻ *Ccr7* KO iNKT and CD45.1⁺ Wt iNKT cells. Yellow outline (middle column) indicates magnified area (right column); Arrows indicate CD45.1⁻ *Ccr7* KO iNKT cells (red), CD45.1⁺ Wt iNKT cells (yellow). C: Cortex; M: Medulla. (E) Flowcytometry analysis of thymocytes (left column), histocytometry analysis of immunofluorescence image (middle column) and localization of CD45.1⁻ *Ccr7* KO iNKT cells (red dot), CD45.1⁺ Wt iNKT cells (yellow dot) as determined by histocytometry (right column). (F) Frequency of iNKT cells localized in thymic medulla from 8-week-old BM chimeras reconstituted with *Ccr7* KO and Wt cells (left column) or with Wt and Wt cells (right column). *$p$=0.0141, *ns*, not significant, $p$=0.1775 (unpaired two tailed $t$ test). Each symbol represents an individual mouse; small horizontal lines indicate the mean.

DOI: https://doi.org/10.7554/eLife.34793.019

The following source data and figure supplement are available for figure 4:

**Source data 1.** CCR7 plays important role in iNKT cells differentiation and localization.
DOI: https://doi.org/10.7554/eLife.34793.021

**Source data 2.** CCR7 plays important role in MAIT cells differentiation.
DOI: https://doi.org/10.7554/eLife.34793.022

**Figure supplement 1.** *Ccr7* deficiency impairs MAIT cells differentiation.
DOI: https://doi.org/10.7554/eLife.34793.020

We also directly tested the retention or emigration of sorted thymic CCR7⁺ iNKT cells (CD4⁺ *Tbx21*ᴳᶠᴾ⁻ CD24⁻ CD8⁻) or *Tbx21*ᴳᶠᴾ⁺ NKT1 (CCR7⁻ CD24⁻ CD8⁻) cells by transferring them intra-thymically into congenic hosts. In contrast to thymic CCR7⁺ iNKT cells, wherein a substantial number of transferred cells were recovered from spleen 5 days after transfer, the transferred *Tbx21*ᴳᶠᴾ⁺ NKT1 cells were rarely recovered from spleen (*Figure 5D,E*), strongly suggest that CCR7⁺ iNKT cells efficiently emigrate thymus while iNKT effector subsets are retained in the thymus. Together, these results show that peripheral iNKT effector subsets are not directly derived from thymic iNKT effector subsets, but rather develop from the CCR7⁺ iNKT progenitors, and indicate that thymic and peripheral pools of iNKT effector subsets do not exchange.

Consistent with their tissue residency, NKT1 and NKT17 cells highly express the classical tissue residency molecules CD69 and CD103, respectively (*Figure 5—figure supplement 1A*). To evaluate whether CD69 and CD103 are required for the thymic retention of NKT1 and NKT17 cells, we analyzed *Cd69*⁺/⁺, *Cd69*⁺/⁻ and *Cd69*⁻/⁻ mice. We observed comparable number of NKT1 cells among these mice in both thymus and spleen (*Figure 5—figure supplement 1B,C*). To block CD103 interaction with E-Cadherin on epithelial cells, monoclonal antibodies (mAb) against CD103 and E-Cadherin were intra-thymically injected three times (*Figure 5—figure supplement 1D,E*). Combined with intra-thymic biotin labeling to track RTE iNKT, we did not observe loss of thymic NKT17 cells nor increase of NKT17 cells in the RTE and peripheral iNKT cells population (*Figure 5—figure supplement 1F*). These data indicate the thymic retention of iNKT effector subsets is unlikely to solely rely on either CD69 or CD103.

That iNKT effector subsets are preferentially retained in the thymus raises the question of their function there. To ask whether NKT1 cells might have effects in the thymus, we investigated if they were producing IFN-γ in the steady state using IFN-γ reporter mice. We observed that ~50% of NKT1 (NK1.1⁺ CD44⁺) cells were *Ifn-γ*ʸᶠᴾ positive (*Figure 5F,G*). Furthermore, Qa2 expression on mature thymocytes was recently shown to be interferon dependent (*Xing et al., 2016*), and was reduced in absence of iNKT cells (*Figure 5H,I*), comparable to that observed in IFN-γ receptor-deficient mice (*Figure 5I*), showing that the IFN-γ produced by NKT1 cells is biologically active, although the functional relevance remains unclear. Lastly, iNKT cells have been shown to regulate mTEC differentiation through the production of RANKL (*White et al., 2014*). We demonstrate here that NKT2 and NKT17 cells are the exclusive producers of RANKL among iNKT cells (*Figure 5J*). Consistent with a previous report (*White et al., 2014*), the absence of iNKT cells resulted in specific reduction of Aire⁺medullary thymic epithelial cells (mTEC), while the number of total TEC was not affected

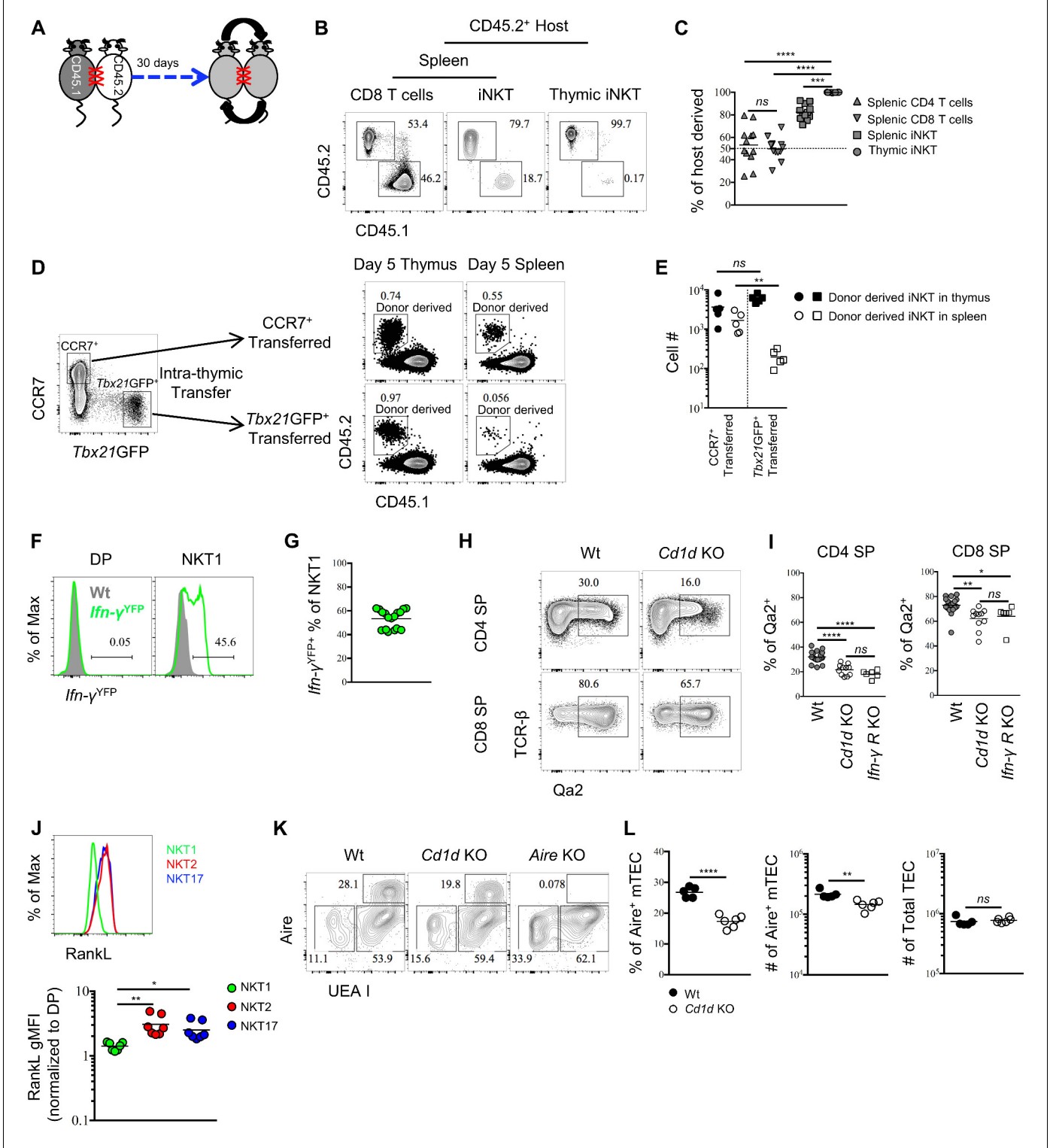

**Figure 5.** Thymic iNKT effector subsets are predominantly resident and may influence thymocytes maturation. (A) Experimental scheme of parabiosis surgery. (B) Representative flowcytometry profile of spleen and thymus from the CD45.2+ host of a parabiotic pair after 30 days of parabiosis surgery, the number indicates the frequency of CD45.2+ (host) and CD45.1+ (parabiotic counterpart) cells in CD8 T cells and iNKT cells. Numbers adjacent to outlined areas indicate percent cells in each throughout. Data are representative of 2 independent experiments with 3 parabiotic pairs in each. (C) The frequency of cells derived from the host parabiont in splenic CD4+ T, CD8+ T and iNKT cells as well as thymic iNKT cells. Data are pooled from 2 independent experiments with 3 parabiotic pairs in each. *ns*, not significant, *p*=0.8722 (one-way ANOVA). ***p*=0.0009, ****p*<0.0001 (one-way ANOVA).
*Figure 5 continued on next page*

*Figure 5 continued*

Each symbol represents an individual mouse; small horizontal lines indicate the mean. (D) CCR7$^+$ *Tbx21*$^{GFP-}$ and CCR7$^-$ *Tbx21*$^{GFP+}$ thymocytes were sorted from BALB/c *Tbx21*$^{GFP}$ mice and intra-thymically transferred to congenic BALB/c recipients (left column). After 5 days, the donor derived iNKT cells were recovered in thymus and spleen from mice received CCR7$^+$ *Tbx21*$^{GFP-}$ thymocytes or CCR7$^-$ *Tbx21*$^{GFP+}$ thymocytes (right column). Numbers adjacent to outlined areas indicate percent cells in each throughout. Data are representative of 5 independent experiments. (E) Cell number of donor derived iNKT cells recovered in thymus and spleen from mice that received CCR7$^+$ *Tbx21*$^{GFP-}$ thymocytes or CCR7$^-$ *Tbx21*$^{GFP+}$ thymocytes. Data are pooled from 5 independent experiments with 2 mice in each. ns, not significant, $p=0.1508$ (unpaired two tailed $t$ test). $^{**}p=0.0079$ (unpaired two tailed $t$ test). Each symbol represents an individual mouse; small horizontal lines indicate the mean. (F). Expression of YFP in DP thymocytes (left column) or thymic NKT1 cells (NK1.1$^+$ CD44$^+$) (right column), the grey shade represents cells from Wt mice, the solid green line represents cells from *Ifn-γ*$^{YFP}$ mice; Numbers adjacent to solid black line indicate percent of *Ifn-γ*$^{YFP+}$ cells from *Ifn-γ*$^{YFP}$ mice; Data are representative of 4 independent experiments with 3–5 mice in each. (G) Frequency of *Ifn-γ*$^{YFP+}$ cells within thymic NKT1 cells. Data are pooled from 4 independent experiments with 3–5 mice in each. Each symbol represents an individual mouse; small horizontal lines indicate the mean. (H) Expression of Qa2 in CD4 SP or CD8 SP thymocytes from Wt or *Cd1d* KO mice. Numbers adjacent to outlined areas indicate percent cells in each throughout. Data are representative of 5 independent experiments. (I) Frequency of Qa2$^+$ cells in CD4 SP or CD8 SP thymocytes from Wt, *Cd1d* KO and *Ifn-γR*$^{KO}$ mice. Data are pooled from 5 independent experiments, n = 18 (Wt), n = 11 (*Cd1d* KO), n = 6 (*Ifn-γR*$^{KO}$). $^*p=0.0426$, $^{**}p=0.0023$, $^{****}p<0.0001$ (one-way ANOVA). ns, not significant, $p=0.2676$ (CD4 SP), $p=0.8848$ (CD8 SP) (one-way ANOVA). Each symbol represents an individual mouse; small horizontal lines indicate the mean. (J) Expression of RankL in thymic NKT1, NKT2 and NKT17 cells (upper panel). Normalized gMFI of RANKL in thymic NKT1, NKT2 and NKT17 cells (bottom panel). Data are representative of 3 independent experiments with 2–3 mice in each. $^{**}p=0.0027$ (NKT1 *vs* NKT2), $^*p=0.0391$ (NKT1 *vs* NKT17) (one-way ANOVA). (K) Expression of Aire in TEC from Wt, *Cd1d* KO or Aire KO mice. Numbers adjacent to outlined areas indicate percent cells in each throughout. Data are representative of 4 independent experiments. (L) Frequency and number of Aire$^+$ mTEC (left and middle column), number of total TEC (right column) from Wt and *Cd1d* KO mice. Data are representative of 4 independent experiments, n = 5 (Wt), n = 6 (*Cd1d* KO). $^{****}p<0.0001$, $^{**}p=0.0032$, ns, not significant, $p=0.5209$ (unpaired two tailed $t$ test). Each symbol represents an individual mouse; small horizontal lines indicate the mean.
DOI: https://doi.org/10.7554/eLife.34793.023

The following source data and figure supplement are available for figure 5:

**Source data 1.** Thymic iNKT cells are resident and influence thymic microenvironment.
DOI: https://doi.org/10.7554/eLife.34793.025
**Source data 2.** CD69 and CD103 are dispensable for thymic retention of NKT1 and NKT17 cells.
DOI: https://doi.org/10.7554/eLife.34793.026
**Figure supplement 1.** NKT1 and NKT17 cells do not rely exclusively on CD69 or CD103 for tissue residency.
DOI: https://doi.org/10.7554/eLife.34793.024

(*Figure 5K,L*). Together these data strongly suggest iNKT cells modulate thymic homeostasis and indicate the possibility that they might further influence T cell development. Given that different inbred strains have strikingly different ratios of iNKT effector subsets in the thymus, our data imply that retention of lipid specific effector T cells in the thymus fine-tunes T cell development and tissue homeostasis.

## Discussion

Previously, PLZF$^{hi}$ iNKT cells have been defined as NKT2 cells, however, our data strongly suggested that CCR7$^+$ PLZF$^{hi}$ iNKT cells had multi-lineage potential in both the thymus and periphery. Thus, PLZF$^{hi}$ iNKT cells comprise a mixture of progenitors and mature IL-4 producing NKT2 effector cells. Such heterogeneity was indicated in the previous RNA-Seq analysis of 'bulk PLZF$^{hi}$ NKT2 cells' (*Lee et al., 2016*; *Engel et al., 2016*). Moreover, our data demonstrate that CCR7 not only marks the precursors but also directs iNKT cells to localize to the thymic medulla, which plays a critical role in their differentiation to effector subsets. This is consistent with a previous report, which showed that *Ccr7* deficient mice harbor fewer thymic iNKT cells (*Cowan et al., 2014*). Considering the pre-dominant medullary localization of all three iNKT effector subsets (*Lee et al., 2015*), it would seem that medullary factors are crucial for the differentiation and/or maintenance/survival of iNKT effector subsets. Indeed, IL-15 derived from medullary thymic epithelial cells (mTEC) was implicated in the generation of T-bet$^+$ NKT1 cells (*White et al., 2014*), (*Cui et al., 2014*). Whether and how medullary factors are required for differentiation of NKT2 and NKT17 cells remains to be determined.

Another type of innate-like T cells, MAIT cells, which are restricted to MR1 and recognize riboflavin metabolites, have gained interest recently. A number of studies indicated a parallel between these iNKT cells and innate-like T cells in their development and functions (*Koay et al., 2016*; *Wang and Hogquist, 2016*; *Rahimpour et al., 2015*). For example, both mature iNKT and MAIT

cells are poised effector cells, producing large amounts of cytokines upon stimulation; both express and depend on PLZF; and both are composed of effector subsets analogous to CD4 T helper subsets. We show here that, similar to iNKT cells, CCR7 also marks a population of immature MAIT cells after the agonist selection. However, whether CCR7 directs their localization remain to be answered.

Mature iNKT effector subsets, including NKT1, NKT2 and NKT17, exist in both the thymus and peripheral tissues (*Lee et al., 2015*). Since iNKT cells originate in the thymus (*Egawa et al., 2005*; *Gapin et al., 2001*), this raises the question whether the peripheral iNKT effector cells directly arise from mature thymic iNKT effector cells or not. Previous work showed that a thymic NK1.1⁻ iNKT cell emigrates (*Benlagha et al., 2002*; *Pellicci et al., 2002*), but the work was performed before it was appreciated that mature NKT2 and NKT17 effector cells are also NK1.1⁻. In this study, we demonstrate that all thymic iNKT effector cells are long-term residents that do not emigrate, and that peripheral iNKT effector subsets derive from CCR7⁺ iNKT cells that emigrate from the thymus. iNKT cells can participate in immunity to infection (*Bendelac et al., 2007*), but they can also regulate tissue homeostasis in the steady state (*Lynch et al., 2012*; *Lee et al., 2015*; *Lynch et al., 2016*; *Lee et al., 2016*). An effect of NKT2 cells on CD8 T cell development in the steady state is well supported by previous data showing their constitutive production of IL-4 drives Eomesodermin (Eomes) expression and a memory-like functional state in CD8 T cells (*Lee et al., 2013*; *Lee et al., 2015*). In this report, we further show that thymic resident NKT1 cells produce IFN-γ and promote Qa2 expression in CD4 and CD8 SP thymocytes. T cells up-regulate Qa2 expression during their maturation in the thymus, and type I interferon signaling has been implicated in this process (*Xing et al., 2016*). It is possible that IFN-γ might function synergistically with type I interferon to influence Qa2 expression. Lastly, we show both NKT2 and NKT17 cells express RANK ligand, and together could promote Aire expression in mTEC in supporting central tolerance.

It is worth mentioning that a small number of CCR7⁺ iNKT cells express lower amounts of PLZF, LEF1 or CD4. Indeed, the development of iNKT cells in the thymus is a dynamic process, even though the CCR7⁺ iNKT cell population is largely immature and maintains precursor potential, it seems likely some of these cells have begun to receive development signals and to differentiate, prior to having completely lost CCR7 expression. Of interest, though CCR7 is critical for optimal differentiation of all three iNKT subsets, NKT17 cells seem to be most affected by CCR7 deficiency. Since NKT1, NKT2 and NKT17 are all localized to the thymic medulla, it is unclear what might cause this effect. Possibly, the differentiation or maintenance of NKT17 cells is more dependent on as yet undefined factors in the medullary environment.

In summary, we demonstrate here that the sub-population of CCR7⁺PLZFʰⁱ iNKT cells represent a pool of precursor cells for all three iNKT effector subsets in thymus and periphery. Meanwhile, CCR7 not only marks the precursors, but also directs the medullary localization and differentiation of iNKT effector subsets. The thymic and peripheral iNKT effector subsets are resident and do not exchange, therefore depend on CCR7⁺ iNKT progenitors for their establishment.

## Materials and methods

**Key resources table**

| Reagent type (species) or resource | Designation | Source or reference | Identifiers | Additional information |
|---|---|---|---|---|
| Antibody | anti-CCR7 | BD Biosciences | Cat # 562675 | (1:50) |
| Antibody | anti-CD4 | BD Biosciences | Cat # 563331 | (1:400) |
| Antibody | anti-CD8α | BD Biosciences | Cat # 563786 | (1:400) |
| Antibody | anti-CD24 | BioLegend | Cat # 101824 | (1:200) |
| Antibody | anti-NK1.1 | BioLegend | Cat # 108718 | (1:100) |
| Antibody | anti-CD44 | TONBO Biosciences | Cat # 80–0441 U025 | (1:200) |
| Antibody | anti-human CD2 | BioLegend | Cat # 309218 | (1:20) |
| Antibody | anti-TCRβ | BD Biosciences | Cat # 563221 | (1:200) |
| Antibody | anti-PD-1 | BioLegend | Cat # 135213 | (1:100) |

*Continued on next page*

*Continued*

| Reagent type (species) or resource | Designation | Source or reference | Identifiers | Additional information |
|---|---|---|---|---|
| Antibody | anti-CD45.1 | BioLegend | Cat # 110738 | (1:200) |
| Antibody | anti-CD45.2 | eBioscience | Cat # 11-0454-81 | (1:200) |
| Antibody | anti-Qa2 | BioLegend | Cat # 121710 | (1:200) |
| Antibody | anti-RANKL | eBioscience | Cat # 12-5952-82 | (1:100) |
| Antibody | anti-B220 | BioLegend | Cat # 103244 | (1:200) |
| Antibody | anti-CD11c | eBioscience | Cat # 47-0114-82 | (1:200) |
| Antibody | anti-CD11b | eBioscience | Cat # 47-0112-82 | (1:200) |
| Antibody | anti-F4/80 | eBioscience | Cat # 47-4801-82 | (1:200) |
| Antibody | anti-CD122 | BD Biosciences | Cat # 562960 | (1:100) |
| Antibody | anti-PLZF | BD Biosciences | Cat # 563490 | (1:200) |
| Antibody | anti-ROR-γt | BD Biosciences | Cat # 562684 | (1:200) |
| Antibody | anti-T-bet | BioLegend | Cat # 644824 | (1:200) |
| Antibody | anti-Egr2 | eBioscience | Cat # 17-6691-80 | (1:100) |
| Antibody | anti-Aire | eBioscience | Cat # 50-5934-82 | (1:100) |
| Antibody | anti-LEF1 | Cell Signaling Technology | Cat # 2230S | (1:100) |
| Antibody | anti-Rabbit IgG AF488 | Invitrogen | Cat # A11034 | (1:400) |
| Antibody | anti-IL-17RB | R and D Systems | Cat # FAB10402G | (1:20) |
| Antibody | Goat-anti-R Phycoerythrin (PE) | Abcam | Cat # ab34721 | (1:200) |
| Antibody | Donkey-anti-Goat AF555 | Abcam | Cat # ab150130 | (1:400) |
| Antibody | Rabbit-anti-β5t | MBL International | Cat # PD021 | (1:100) |
| Antibody | Goat-anti-Rabbit BV480 | BD biosciences | Cat # 564879 | (1:400) |
| Antibody | anti-EpCAM | BioLegend | Cat # 118214 | (1:200) |
| Other | streptavidin-PE | BD biosciences | Cat # 554061 | (1:400) |
| Other | streptavidin-BV421 | BioLegend | Cat # 405225 | (1:400) |
| Other | Ulex Europaeus Agglutinin I (UEA I) | VECTOR LOBORATORY | FL-1061 | (1:200) |
| Commercial assay or kit | Viability dye Ghost Dye Red 780 | TONBO Biosciences | Cat # 13–0865 T100 | (1:500) |
| Commercial assay or kit | anti-biotin MACS beads | Miltenyi | Cat # 130-105-637 | |
| Commercial assay or kit | anti-PE microbeads | Miltenyi | Cat # 130-048-801 | |
| Chemical compound, drug | Bulsulfan | MP Biomedicals | Cat # 154906 | |
| Chemical compound, drug | Streptavidin | Jackson Immuno Research | Cat # 016-000-113 | |
| Chemical compound, drug | DNase I | Roche | Cat # 10104159001 | |
| Chemical compound, drug | Liberase TH | Roche | Cat # 5401127001 | |
| Chemical compound, drug | sulfo-NHS-LC biotin | ThermoFisher Scientific | Cat # 21335 | |
| Software, algorithm | SPADE | CytoBank | https://www.cytobank.org/ | |

*Continued on next page*

*Continued*

| Reagent type (species) or resource | Designation | Source or reference | Identifiers | Additional information |
|---|---|---|---|---|
| Software, algorithm | Prism 7 | GraphPad | https://www.graphpad.com/ | |
| Software, algorithm | ImageJ | ImageJ | https://imagej.nih.gov/ij/ | |
| Software, algorithm | FlowJo v10 | TreeStar Flowjo | https://www.flowjo.com/solutions/flowjo | |

## Mice

B6 (C57BL/6NCr) and B6.SJL (B6-LY5.2/Cr) mice were purchased from the National Cancer Institute. BALB/c (BALB/cJ or BALB/cBYJ as indicated), congenic CD45.1$^+$ BALB/cBYJ (CByJ.SJL(B6)-Ptprca/J), CD4$^{Cre}$ B6, *Ifn-γ*$^{YFP}$ B6 (B6.129S4-Ifng$^{tm3.1Lky/J}$), *Ifn-γR*$^{KO}$ B6 (B6.129S7-Ifngr1$^{tm1Agt/J}$), *Aire*$^{KO}$ B6 (B6.129S2-Aire$^{tm1.1Doi/J}$), *Ccr7*$^{KO}$ B6 (B6.129P2(C)-Ccr7$^{tm1Rfor/J}$), *Cd1d*$^{-/-}$ B6 (B6.129S6-Cd1d1/Cd1d2$^{tm1Spb/J}$) and *Cd1d*$^{-/-}$ BALB/c.J (C.129S2-Cd1tm1Gru/J) mice were obtained from the Jackson Laboratory. *Rag2*$^{GFP}$ B6, *Klf2*$^{GFP}$ B6, *Klf2*$^{fl/fl}$ B6, KN2 BALB/cBYJ and B6, *Tbx21*$^{GFP}$ Balb/cBYJ and *Tbx21*$^{GFP}$ KN2 BALB/cBYJ have been previously described (*Lee et al., 2013*; *Lee et al., 2015*; *McCaughtry et al., 2007*; *Skon et al., 2013*; *Weinreich et al., 2009*). *Cd69*$^{-/-}$ B6 (*Cd69* KO) mice were kindly provided by Dr. Linda Cauley at University of Connecticut Health Center. CD4$^{Cre}$*Klf2*$^{fl/fl}$ (*Klf2* cKO) mice were generated through crossbreeding at University of Minnesota. All animal work was conducted in compliance with the protocols approved by the Institutional Animal Care and Use Committee of the University of Minnesota.

## Flow cytometry and antibodies

Single-cell suspensions were prepared from thymus and spleen. Biotinylated PBS57 loaded or unloaded CD1d monomers and MR1–5–OP–RU or control MR1–Ac–6–FP monomers were obtained from the tetramer core of the US National Institutes of Health and tetramerized with streptavidin-PE (invitrogen) or streptavidin-APC (invitrogen) at the ratio of 4:1. For staining of CCR7, cells were stained with antibody to CCR7 (4B12, BD Biosciences) at 37°C for 45 min. For intracellular staining of transcription factors, cells were stained with antibodies to surface makers, then fixed and permealized with a Foxp3 staining buffer set (eBioscience), and were incubated with antibodies to PLZF (R17-809, BD Biosciences), T-bet (4B10, Biolegend), ROR-γt (Q31-378, BD Biosciences), Egr2 (erongr2, eBioscience) or Aire (5H12, eBioscience). For staining of LEF1, fixed and permealized cells were incubated with antibody to LEF1 (C12A5, Cell Signaling Technology), followed by staining with secondary antibody to Rabbit IgG (A11034, Invitrogen). Antibody to mouse IL-17RB was from R and D Systems (752101, rat IgG1). Ulex Europaeus Agglutinin I (UEA I) was from VECTOR LOBORATORY. Viability dye Ghost Dye Red 780 was from TONBO Biosciences. The complete list of other antibodies used is as follows and were purchased from eBioscience, BD Biosciences or BioLegend, unless otherwise indicated: anti-CD4 (GK1.5 or RM4-4), CD8α (53–6.7), anti-CD24 (M1/69), anti-NK1.1 (PK136), anti-CD44 (IM7), anti-human CD2 (TS1/8 or RPA-2.10), anti-TCRβ (H57-597), anti-PD-1 (29F.1A12), anti-CD45.1 (A20), anti-CD45.2 (104), anti-Qa2 (695H1-9-9), anti-RANKL (IK22/5), anti-B220 (RA3-6B2), anti-CD11c (N418), anti-CD11b (M1/70), anti-F4/80 (BM8), anti-CD122 (TM-β1). Cells were analyzed on a BD LSR Fortessa or BD Fortessa X-20 and data were processed with FlowJo software.

## Enrichment of iNKT cells or MAIT cells

Single cell suspensions were prepared from thymus or spleen and incubated with PE-CD1d-PBS57 tetramer at 4°C or with PE-MR1–5–OP–RU tetramer at room temperature. Then anti-PE microbeads (Miltenyi) were used for immunomagnetic enrichment according to the manufacturer's instructions.

## SPADE analysis of thymic iNKT cells

Thymic iNKT cells were enriched from B6 KN2 or BALB/c KN2 thymus. Enriched iNKT cells were stained with antibodies to CCR7, TCR-β, IL-17RB, CD1d tetramer, CD24, T-bet, PD-1, PLZF, CD44, Dump (viability dye, B220, F4/80, CD11c), CD8, ROR-γt, human CD2, CD4. Total thymic iNKT cells

(TCR-$\beta^+$ CD1d tetramer$^+$ CD24$^-$) were subjected to SPADE analysis on Cytobank (www.cytobank.org) based on expression of following surface markers, key transcription factors and IL-4 production: CCR7, IL-17RB, PD-1, CD44, CD4, T-bet, PLZF, ROR-$\gamma$t, human CD2.

## Intra-thymic and intravenous injection

Thymocytes from BALB/c KN2, BALB/c *Tbx21*$^{GFP}$ or BALB/c *Tbx21*$^{GFP}$ KN2 mice were depleted of CD8$^+$ and CD24$^+$ cells (via immunomagnetic selection [Miltenyi]) to enrich thymic iNKT cells. CCR7$^+$ CD4$^+$ human CD2$^-$, CCR7$^+$ *Tbx21*$^{GFP-}$, CCR7$^+$ CD4$^+$ *Tbx21*$^{GFP-}$ human CD2$^-$ cells or CCR7$^-$ *Tbx21*$^{GFP+}$ cells were sorted with a FACSAria. Ultrasound imaging guided intra-thymic injections were performed on congenic host mice (BALB/c) using a Vevo 2100 preclinical scanner (VisualSonics) with a MS550 transducer. Ultrasound guided intrathymic injection is described in detail at Bioprotocol (*Wang et al., 2018*). Mice received intra-thymic transfer or intravenous injections of cells were analyzed 5 days after injections. For intra-thymic injection of biotin to track RTEs, mice were given ~10 µL of 1 mg/mL solution of sulfo-NHS-LC biotin. At 24 hr or 74 hr later, biotin$^+$ RTEs in spleen were enriched by immunomagnetic selection (via anti-biotin MACS beads [Miltenyi]), and stained with streptavidin-PE or streptavidin-BV421, followed by blocking with free streptavidin (Jackson ImmunoResearch) before staining for surface markers and CD1d tetramer.

## Bone marrow chimera

Bone marrow cells were prepared from the femurs and tibias of donor mice, and depleted of T cells with CD90.2 MACS beads. Recipient mice (CD45.1$^+$ CD45.1$^+$) were lethally irradiated (2 × 500 rads) and received 5 × 10$^6$ T-cell-depleted bone marrow cells. Mixed donor cells consisted of 25% of *Klf2* cKO cells (CD45.2$^+$ CD45.2$^+$) and 75% of Wt cells (CD45.1$^+$ CD45.2$^+$) or 50% of *Ccr7* KO cells (CD45.2$^+$ CD45.2$^+$) and 50% of Wt cells (CD45.1$^+$ CD45.2$^+$) or 50% of Wt cells (CD45.2$^+$ CD45.2$^+$) and 50% of Wt cells (CD45.1$^+$ CD45.2$^+$). All chimeras were analyzed 7 or 8 weeks after transplantation unless otherwise indicated. For busulfan induced bone marrow chimeras, recipient mice (CD45.2$^+$ CD45.2$^+$) were injected intraperitoneal with 400 µg of busulfan on two consecutive days, and received 1 × 10$^7$ T-cell-depleted bone marrow cells (CD45.1$^+$ CD45.1$^+$). Busulfan induced bone marrow chimeras were analyzed at 4 and 5 weeks after transplantation.

## Parabiosis surgery

Parabiosis surgeries were performed as previously described (*Skon et al., 2013*). Briefly, mice were anesthetized with ketamine, and flank hair was shaved and further removed using Nair. After making lateral incisions, mice were joined with interrupted horizontal mattress sutures with 5–0 NOVAFIL. Additional sutures were placed through the olecranon and knee joints to secure the legs. Parabiotic pairs were analyzed 30 days after surgeries.

## Preparation of thymic epithelial cells

The procedure for preparation of thymic epithelial cells have been previously described (*Xing and Hogquist, 2014*). Briefly, fresh thymus was digested with 0.05% [w/v] of Liberase TH (Roche) and 100 U/ml of DNase I (Roche) and mechanically disrupted. After complete digestion, flowcytometic analysis was performed, and total thymic epithelial cells were identified as CD45$^-$ EpCAM$^+$ cells.

## Immunofluorescence

The CD1d tetramer immunofluorescence has been described previously (*Lee et al., 2015*). Briefly, fresh thymic lobes were incubated with PE-CD1d-PBS57 tetramer at 4°C overnight. The tissues were then washed with PBS and fixed in 4% paraformaldehyde (PFA) for 1 hr and snap frozen in OCT. Five micrometer sections were made and stained with Goat-anti-R Phycoerythrin (PE) (Abcam) followed by Donkey-anti-Goat AF555 (Abcam). The sections were incubated with Rabbit-anti-β5t (MBL International) and anti-CD45.1 AF647 (Biolegend) at 4°C overnight, followed by Goat-anti-Rabbit BV480 (BD biosciences). The sections were counterstained with DAPI covered with Prolong anti-fade mounting medium (Life Technologies). The images were obtained with a Leica DM6000B Epi-Fluorescent microscope.

## Histocytometry

The histocytometry analysis was described previously (*Lee et al., 2015*; *Gerner et al., 2012*). Briefly, the region of interests (ROIs) were identified and the fluorochrome intensities of each ROI were quantified using ImageJ and data were exported into Excel, Prism and FlowJo software for the localization analysis.

## Ultrasound imaging guided intra-thymic injection

The intra-thymic injection was guided by ultrasound imaging using the Vevo 2100 preclinical scanner (VisualSonics) with the MS550 transducer, which ranges from 32 to 40 MHz. Briefly, we identified the thymus in the ultrasound imaging, then adjusted and slid the needle under the transducer until it could be clearly visualized above the skin of the chest area. The needle was then inserted into the thymus gland, approaching continuously toward the thymus area while avoiding penetrating any attached blood vessels or going underneath the aorta. Once the needle tip was observed within the thymic area, a 10 µL volume was injected.

## Statistical analysis

Prism software (GraphPad) was used for statistical analysis. Unpaired or paired two-tailed t-tests, and one-way ANOVA were used for data analysis and calculation of $p$ values. Data sets (in Prism format) are available per request.

## Acknowledgements

We thank Dr. Jason Schenkel (current affiliation: Brigham and Women's Hospital) and Ms. Jane Ding for technical assistance, Drs. Stephen Jameson and You Jeong Lee (current affiliation: Pohang University of Science and Technology (POSTECH), Korea) for discussions and technical comments, and Dr. Henrique Borges da Silva and Mr. Dmitri Kotov for reviewing the manuscript. This research was supported by NIH grant R37 AI39560.

## Additional information

### Funding

| Funder | Grant reference number | Author |
| --- | --- | --- |
| National Institute of Allergy and Infectious Diseases | R37 AI39560 | Kristin A Hogquist |

The funders had no role in study design, data collection and interpretation, or the decision to submit the work for publication.

### Author contributions

Haiguang Wang, Conceptualization, Investigation, Methodology, Writing—original draft, Writing—review and editing; Kristin A Hogquist, Conceptualization, Supervision, Funding acquisition, Project administration, Writing—review and editing

### Author ORCIDs

Haiguang Wang (iD) http://orcid.org/0000-0001-5403-9179
Kristin A Hogquist (iD) http://orcid.org/0000-0001-9963-5687

### Ethics

Animal experimentation: This study was performed in strict accordance with the recommendations in the Guide for the Care and Use of Laboratory Animals of the National Institutes of Health. This protocol was approved by the Institutional Animal Care and Use Committee of the University of Minnesota (protocol #s1706-34889A and 1502-32279A).

Decision letter and Author response
Decision letter https://doi.org/10.7554/eLife.34793.029
Author response https://doi.org/10.7554/eLife.34793.030

## Additional files

### Supplementary files
• Transparent reporting form
DOI: https://doi.org/10.7554/eLife.34793.027

### Data availability

All relevant data are included in the manuscripts and supporting files. Source data files have been provided.

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
