## [Decision Letter]

Thank you for submitting your article "CCR7 defines a precursor for iNKT cells in thymus and periphery" for consideration by *eLife*. Your article has been reviewed Arup Chakraborty as the Senior Editor, a Reviewing Editor, and two reviewers. The reviewers have opted to remain anonymous.

The reviewers have discussed the reviews with one another and the Reviewing Editor has drafted this decision to help you prepare a revised submission.

Summary:

In this manuscript, the authors demonstrate that CCR7 is expressed by a subset of CD1d tet^+^ iNKT cells in the thymus that correspond to the previous stage 1 cells. The CCR7^+^ iNKT cells, when injected intrathymically, give rise to cells that express markers of all three iNKT lineages. A small amount of data is also provided for MAIT cells suggesting that precursors of these cells are also CCR7^+^. Furthermore, CCR7^+^ iNKT cells emigrate to the spleen in a KLF2-dependent manner. After intrathymic injection, CCR7^+^ cells give rise to all three effector iNKT subsets in the spleen. Because parabiosis experiments demonstrate little recirculation of peripheral iNKT effector cells, the authors suggest thymic CCR7^+^ iNKT cells migrate into peripheral organs and differentiate into tissue-resident cells therein. In the thymus, the effector iNKT cells express cytokines and impact differentiation of Aire^+^ mTEC. CCR7 promotes entry of iNKT into the medulla and is required for their intrathymic differentiation. Altogether, identification of CCR7^+^ iNKT progenitors is an important advance that will enable further dissection of the mechanisms governing intrathymic and peripheral iNKT differentiation. However, there is some additional data and changes to the manuscript that will be important to clarify the impact of these findings.

Note: There are several grammatical and typographical problems that should be checked throughout the manuscript.

Essential revisions:

1) The highest priority for the revision is to demonstrate that CCR7^+^ precursors can really generate the three separate lineages. The model put forward is that CCR7^+^ iNKT in the thymus are progenitors that give rise to all 3 effector iNKT subsets in the thymus and emigrate to the periphery where they can also differentiate, at least to NKT1 T-bet^+^ cells, in the spleen. To support the claim that CCR7^+^ progenitors emigrate from the thymus and differentiate in the periphery, it should be shown that if you intravenously inject CCR7^+^ progenitors into a congenic recipient, they give rise to NKT2 and NKT17, as well as NKT1 (as in Figure 1—figure supplement 1F). In this case, the spleen may not be the right organ to examine since at least ROR-γt^+^ iNKT are largely absent in the spleen (Figure 3A), but perhaps these iNKT types could be found in other tissues? However, in Figure 3—figure supplement 1, the authors do see ROR-γt^+^ and IL-4^+^ (huCD2^+^) cells in the spleen. Please explain this discrepancy. It is important to have a direct demonstration that the CCR7^+^ iNKT progenitors differentiate into all 3 iNKT subsets in the periphery to support the overall conclusion of the paper.

2) The rationale for the SPADE analysis was to identify markers that could be used to gate on iNKT precursor cells: were any markers beyond the known CD1d tet^+^ CD44^-^ NK1.1^-^ CCR7^+^ CD24^-^ identified? Figure 1H doesn't provide additional markers.

3) The statistical tests are not correctly applied. For example, in Figure 1D, a t-test is used to compare multiple groups to each other. Unless this data is corrected for multiple hypothesis testing, the p values will be misleading. The same applies for many other figures in the paper where more than 1 comparison is made per experiment. ANOVA might be more appropriate for these tests. ANOVA is mentioned in the methods section but not in the legends. Also, the line in each scatter graph presumably represents the mean, but this should be stated.

4) The result identifying CCR7^+^ MAIT cell precursors was interesting and given that the authors are able to identify these cells, it is not clear why MAIT cells were not also examined in other parts of the paper. What was the transcription factor status of the CCR7 MAIT cells (PLZF, T-bet, ROR-γt)? Was MAIT cell differentiation impaired in *Ccr7* KO mice?

---

## [Author Response]

Essential revisions:1) The highest priority for the revision is to demonstrate that CCR7^+^ precursors can really generate the three separate lineages. The model put forward is that CCR7^+^ iNKT in the thymus are progenitors that give rise to all 3 effector iNKT subsets in the thymus, and emigrate to the periphery where they can also differentiate, at least to NKT1 T-bet^+^ cells in the spleen. To support the claim that CCR7^+^ progenitors emigrate from the thymus and differentiate in the periphery, it should be shown that if you intravenously inject CCR7^+^ progenitors into a congenic recipient, they give rise to NKT2 and NKT17, as well as NKT1 (as in Figure 1—figure supplement 1F). It is important to have a direct demonstration that the CCR7^+^ iNKT progenitors differentiate into all 3 iNKT subsets in the periphery to support the overall conclusion of the paper.

We agree and now provide this new data in Figure 1—figure supplement 2F and G. Upon intravenous injection of CCR7^+^ progenitors, we observed differentiation into NKT1, NKT2 and NKT17 cells in the spleen 5 days later. We failed to recover a sufficient number of transferred cells from skin draining or mesenteric LN for analysis.

In this case, the spleen may not be the right organ to examine since at least ROR-γt^+^ iNKT are largely absent in the spleen (Figure 3A), but perhaps these iNKT types could be found in other tissues? However, in Figure 3—figure supplement 1, the authors do see ROR-γt^+^ and IL-4^+^ (huCD2^+^) cells in the spleen. Please explain this discrepancy.

There is not a big discrepancy here, although we note that the different Y-axis scales in Figure 3 and Figure 3—figure supplement 1 make them appear strikingly different. The frequency of splenic ROR-γt^+^ NKT17 cells shown in Figure 3—figure supplement 1 is 3.3 ± 2.2 whereas it is either 2.8 ± 1.4 or 1.34 ± 0.7 in Figure 3, depending on whether gated on RTE or non-RTE. The frequency and number of cells are also provided in the source data file.

2) The rationale for the SPADE analysis was to identify markers that could be used to gate on iNKT precursor cells: were any markers beyond the known CD1d tet^+^ CD44^-^ NK1.1^-^ CCR7^+^ CD24^-^ identified? Figure 1H doesn't provide additional markers.

No, we did not identify other cell surface proteins that were different, for which antibodies are available. Antibodies to CCR7 and PD-1 best distinguish progenitors from IL-4 producing NKT2 cells within the PLZF^hi^ iNKT population. This is now noted in the text subsection “CCR7^+^ iNKT and MAIT cells are at an early stage of development and represent a precursor pool for effector subsets in the thymus”.

3) The statistical tests are not correctly applied. For example, in Figure 1D, a t-test is used to compare multiple groups to each other. Unless this data is corrected for multiple hypothesis testing, the p values will be misleading. The same applies for many other figures in the paper where more than 1 comparison is made per experiment. ANOVA might be more appropriate for these tests. ANOVA is mentioned in the methods section but not in the legends. Also, the line in each scatter graph presumably represents the mean, but this should be stated.

ANOVA was used in statistical analysis where multiple comparisons were involved in Figure 1D, Figure 2B, Figure 2D, Figure 3B, Figure 3D, Figure 5C, Figure 5I, Figure 5J, Figure 1—figure supplement 3C, and Figure 5—figure supplement 1C. None of the conclusions we reached in earlier version of the manuscript were changed. The small bar in scatter graphs represents the mean, which is now included in the figure legends.

*4) The result identifying CCR7^+^ MAIT cell precursors was interesting and given that the authors are able to identify these cells, it is not clear why MAIT cells were not also examined in other parts of the paper. What was the transcription factor status of the CCR7 MAIT cells (PLZF, T-bet, ROR-γt)? Was MAIT cell differentiation impaired in Ccr7 KO mice?*

We performed experiments to enrich thymic MAIT cells using MR1/5-OP-RU tetramers, and stained for surface CCR7 together with intracellular transcription factors. Consistent with a previous study (Koay et al., 2016), mature MAIT cells were composed of only two subsets, ROR-γt^+^ MAIT cells and T-bet^+^ MAIT cells. We show here that CCR7^+^ MAIT cells express neither T-bet or ROR-γt (Figure 1—figure supplement 1E). We also generated *Ccr7* KO and wild-type mixed BM chimeras to check whether MAIT cell differentiation was influenced by CCR7. Consistent with what we reported for iNKT cells, we observed a severe defect in the subset differentiation of MAIT cells (much less ROR-γt^+^ and T-bet^+^ MAIT cells) in the *Ccr7* KO cells (Figure 4—figure supplement 1).